# Identifying Risk Factors for Lower Reproductive Tract Infections among Women Using Reusable Absorbents in Odisha, India

**DOI:** 10.3390/ijerph18094778

**Published:** 2021-04-29

**Authors:** Padmalaya Das, Danielle Lisnek, Krushna Chandra Sahoo, Shalini Sinha, JyotiRanjan Mohanty, Pranati Sahoo, Bibiana Bilung, Bijaya Panda, Clare Tanton, Belen Torondel

**Affiliations:** 1School of Biological Sciences, AIPH University, Bhubaneswar 752101, Odisha, India; pdas1234@gmail.com (P.D.); shalini.sinha58@yahoo.com (S.S.); pranatip123@gmail.com (P.S.); 2London School of Hygiene and Tropical Medicine, London WC1E 7HT, UK; daniellelisnek@gmail.com (D.L.); clare.tanton@lshtm.ac.uk (C.T.); 3ICMR-Regional Medical Research Centre, Bhubaneswar 751023, Odisha, India; sahookrushna@yahoo.com; 4Disease Surveillance Laboratory, Asian Institute of Public Health, Bhubaneswar 751002, Odisha, India; JRrmmohanty@aiph.ac.in; 5Department of Obstetrics and Gynecology, Ispat General Hospital, Rourkela 769005, Odisha, India; drpiabilung@gmail.com; 6Department of Obstetrics and Gynecology, Capital Hospital, Bhubaneswar 751001, Odisha, India; bijayap662@gmail.com

**Keywords:** menstrual hygiene management, reusable absorbents, bacterial vaginosis, candidiasis, reproductive tract infections

## Abstract

A large proportion of women in Odisha, India, use reusable absorbents to manage their menstruation. Yet, the risk factors for lower reproductive tract infections (RTIs) related to menstrual hygiene management (MHM) have not been studied among reusable absorbent users. Women of reproductive age attending one of two hospitals from two different cities in Odisha during two separate study intervals were recruited for the study. Laboratory diagnosis of bacterial vaginosis (BV) and vulvovaginal candidiasis (VVC) were conducted. A questionnaire was used to collect information on MHM practices, water, sanitation, and socio-demographic factors. Among the 509 women who used reusable absorbents, 71.7% were diagnosed with at least one infection. After adjusting for confounders, women with BV were more likely to identify as being a housewife (aOR: 1.8 (1.1–2.9)). Frequent absorbent changing was protective against BV (aOR: 0.5 (0.3–0.8)), whereas frequent body washing increased the odds of BV (aOR: 1.5 (1.0–2.2)). Women with VVC were more likely to be older (aOR: 1.6 (1.0–2.5)), live below the poverty line (aOR: 1.5 (1.1–2.2)), have a non-private household latrine (aOR: 2.2 (1.3–4.0)), dry their absorbents inside the house (aOR: 3.7 (2.5–4.5)), and store absorbents in the latrine area (aOR: 2.0 (1.3–2.9)). Washing absorbents outside the house was protective against VVC (aOR: 0.7 (0.4–1.0)). This study highlights the importance of improving MHM practices among reusable absorbent users to prevent lower RTIs among women reusing menstrual materials in Odisha.

## 1. Introduction

Menstruation is a natural, healthy, and frequent occurrence in the life cycle of most women and adolescent girls. However, women and girls face significant challenges to managing their menstruation, especially in low- and middle-income countries (LMICs) [1,2]. Menstrual hygiene management (MHM) practices vary between and within countries and are dependent on the materials and resources accessible, access to water, sanitation and hygiene (WASH), and socio-economic status (SES) [1,3]. Local traditions, education, and cultural and religious beliefs also influence MHM practices [1,3]. These restrictions often drive women and girls to turn to unhygienic, inconvenient, and uncomfortable methods to manage their menstruation, particularly in LMICs [1]. Materials typically used as absorbents in LMICs are fabric-based cloths and rags, which are washed and reused, although many girls and women utilize toilet paper, gauze, newspaper, mattress stuffing, dry leaves, grass, or cow dung [4,5,6]. Evidence suggests that poor MHM practices are associated with psychosocial stress rooted in feelings of shame, fear of stigma, anxiety, distraction, and disengagement [5,7,8,9,10]. Additionally, unhygienic MHM practices related to the use, washing, drying, and storage of menstrual products have been shown to be linked with poor health outcomes, such as urogenital infections [6,7,8].

Urogenital infections, including reproductive tract infections (RTIs), are a major public health concern, which are particularly common in LMICs [7]. The most common RTI is bacterial vaginosis (BV), characterized by the imbalance of bacterial flora in the vagina. In BVpositive patients, there is a decline in *Lactobacillus* colonization (such as *Lactobacillus crispatus*, *Lactobacillus jensenii*, and *Lactobacillus iners*) with a simultaneous increase in facultative anaerobic bacteria in the vaginal microbiome (such as *Gardnerella vaginalis*, *Prevotella bivia*, *BV-associated-bacteria-2 (BVAB2)*, *Megasphaera 1*, and *Atopobium vaginae*) [7,11]. Symptoms of BV include vaginal irritation, pruritus, burning while urinating, and discharge with abnormal odor, color, and consistency, though as many as 50% of women are asymptomatic [7,11,12]. BV is particularly a concern for women of reproductive age as it is associated with increased risk of sexually transmitted infections (STIs) [13], increased risk of HIV-1 infection [14], pelvic inflammatory disease (PID) [7,11,15], and adverse pregnancy outcomes, such as miscarriage, preterm labor, preterm delivery, and postpartum complications [16,17,18]. The second most common RTI is *Candida* infection. Vulvovaginal candidiasis (VVC), the disease state of *Candida* infection, affects up to 75% of reproductive-age women at least once over their lifetime, with 40% to 45% of women experiencing two or more episodes [7,19]. VVC is most commonly caused by *Candida albicans* but can also be caused by other *Candida* species or yeasts [19]. Symptoms of VVC include pruritus, vaginal soreness, dyspareunia, external dysuria, and abnormal vaginal discharge [19,20,21]. Approximately 10% to 20% of women with *Candida albicans* present in the vagina are asymptomatic [19]. Both BV and VVC are curable given timely and proper treatment. However, social, cultural, psychological, and economic barriers hinder health-seeking behavior for many women, especially in LMICs [22,23]. Although prevention and control of RTIs have been accorded a national priority in India, they remain an increasingly prevalent public health issue [22]. As reported in the District Level Household Survey—Reproductive and Child Health (DLHS-RCH), the prevalence of RTIs and STIs among women of reproductive age in Odisha was 35.2% in 2002–2004, a 126% increase from 1998–1999 [22]. The prevalence of BV and VVC specifically in India are unknown, as there is a lack of population-based prevalence surveys conducted. Estimates from Obstetrics and Gynecology (O&G) departments and medical clinics most likely underestimate the true burden of disease, as a high proportion of cases are asymptomatic and thus unreported [24,25].

The majority of studies exploring menstrual product use and RTIs have utilized self-reported symptoms to diagnose RTIs, which may result in biased effect estimates due to the high prevalence of asymptomatic infections and the non-exclusivity of the risk factors for a range of RTIs [1,7,25]. Two studies in Odisha utilized laboratory diagnosis of RTIs and found an association between poor MHM practices and BV and VVC, with an increased risk of infection among women who used reusable absorbents compared to disposable pads [7,25]. Disposable pads may address some of the issues associated with poor MHM practices related to reusable pads, however, they are often inaccessible in LMICs due to the high cost and scarcity of products. They also present challenges in terms of environmental impact and sustainability of sanitation systems. In India, between 43% and 88% of women and girls wash and reuse absorbents made from old cloths or saris, especially in rural areas [1,25]. Thus, it is essential that women and girls are empowered with information about how to manage their menstruation safely and with dignity, particularly when using reusable absorbents. Given the high proportion of women who use reusable absorbents in Odisha (51.2%) [25], more research is needed to identify specific MHM practices that increase their risk of infection. 

In this study, we aim to address this gap by implementing the following objectives: (1) explore reusable menstrual product use, MHM practices, and WASH access among reproductive-aged women in Odisha; and (2) examine the specific risk factors related to MHM practices among reproductive-aged women in Odisha who use reusable menstrual products and their associations with the most common RTIs: BV and VVC. 

## 2. Materials and Methods 

### 2.1. Study Design and Study Population

This study is a combination of two hospital-based cross-sectional studies, referred to as Study One and Study Two. Non-pregnant women of reproductive age attending the O&G outpatient department (OPD) with vaginal symptoms, including vaginal discharge, itching, burning, or dyspareunia; with lower abdominal or lower back pain; or attending the Family Welfare Department, were eligible to enroll in the studies. Study One recruited women attending either Capital Hospital in Bhubaneswar or Ispat General Hospital (IGH) in Rourkela between April 2015 and February 2016. Study Two recruited women attending Capital hospital between June 2017 and March 2018. Capital Hospital is a government hospital with the majority of patients from urban or peri-urban slums in Bhubaneswar city as well as adjoining rural areas. It has approximately 700 beds and specializes in O&G and family welfare with the majority of health services free of cost. IGH is managed by the Steel Authority of India Limited and provides most services at subsidized costs. The majority of patients at IGH are from tribal populations in Rourkela.

Due to the known increased risks for lower RTIs, both studies excluded women meeting one or more of the following criteria: menstruating during the clinic visit, had a hysterectomy, taken a course of antibiotics during the previous three weeks, used oral contraceptive pills in the previous three months, had diabetes mellitus, or were HIV positive. Women were also excluded if they had any severe medical disorders requiring immediate referral to a higher level of health care. Only women who used reusable absorbents were included in the analysis.

Among women who provided consent and fit the criteria for the study, pen and paper standardized questionnaires were used to collect information on socio-demographic and economic factors, clinical symptoms associated with BV and VVC, MHM practices, body hygiene habits, and the WASH conditions in their households (Appendix A). Questionnaires were administered by trained female interviewers in a private room. The questionnaires used for the two studies were slightly different and only questions that were asked in both studies were included in the analysis. Some variables were recoded for consistency between studies. Specimens from the posterior vaginal fornix were collected using swabs for laboratory diagnosis of BV and VVC. 

### 2.2. Diagnosis of Outcomes: BV and VVC

The primary outcomes for this study are the lab-confirmed presence of BV or VVC. At the point of data collection in the hospital, women underwent a vaginal and cervical speculum examination performed by a trained O&G specialist. The presence of cervical erythema, bleeding, inflammation, and cervical ulcers were assessed. Vaginal specimens from the posterior vaginal fornix were collected using four BD BBL Culture Swabs (BD, Sparks, MD and USA). The first swab was used for Gram staining to diagnose BV. Nugent’s laboratory diagnostic criteria was used to diagnose BV and was performed by trained personnel and a Nugent score (NS) was generated using the Nugent criteria [26]. The Nugent score was calculated by assessing for the presence of large Gram-positive rods (*Lactobacillus* morphotypes; decrease in *Lactobacillus* scored as 0 to 4), small Gram-negative to Gram-variable rods (*G.vaginalis* morphotypes, scored as 0 to 4), and curved Gram-variable rods (*Mobiluncus* spp. morphotypes, scored as 0 to 2); the score is generated from combining the three scores and can range from 0 to 10. A score of 0–3 is considered BV negative, 4–7 BV intermediate, and 7–10 BV positive. For this study an NS of 0–3 was considered negative for BV, while a score of 4–10 considered positive for BV. Patients with a positive BV score were referred to further treatment in the respective hospitals. The second swab was used to diagnose VVC by identifying the presence of *Candida albicans* using the AlbiQuickTM rapid test (HARDY Diagnostic, Santa Maria, CA, USA) [26]. The third swab was used for diagnosis of *Trichomonas vaginalis* (TV). The fourth swab was stored for future studies. For the purpose of this study, TV diagnoses were excluded and only the BV and VVC results were analyzed.

### 2.3. Identification of Risk Factors

Risk factors for BV and VVC were divided into three groups: socio-demographic characteristics, WASH access variables, and MHM practices. Information on each risk factor was explored through standardized questionnaires. Socio-demographic factors included the hospital attended, age, number of family members in the household, socio-economic status (SES), education, occupation, religion, and marital status. Age was treated as a categorical variable and divided into three age groups: 18–25, 26–35, and 36–45. Education was measured as the highest level of education completed, divided into three categories: no formal education, some formal education, including primary or secondary school, and any level of higher education. Ownership of a Below Poverty Line (BPL) card, which is given to women who fall below the poverty line in India, was used as a proxy for SES. WASH access variables included where women urinate during menstruation, the location of the primary water source, and household latrine privacy. Household latrine privacy was derived from the combination of household latrine access and privacy, defined by the respondent’s perceived feeling that they have enough privacy at their primary latrine. Household latrine privacy was divided into three categories: no household latrine, a household latrine without privacy (non-private HH latrine), and a household latrine with privacy (private HH latrine). MHM practices were divided into four sub-groups. Practices related to absorbent changing and body washing included the primary location and frequency of absorbent changing as well as the usual type and frequency of body washing during menstruation. The type of absorbent material pertains to the material most commonly used within the past six menstrual cycles. Washing of reusable absorbents included the most common location and method of absorbent washing. Practices related to drying and storage included the location where women most commonly dry their absorbents as well as the location and method of storing absorbents between cycles. 

### 2.4. Statistical Analysis

All data were doubleentered into Epi Info 7 software (Epi info, Centers for Disease Control and Prevention (CDC), Atlanta, GA, USA) and analyzed using Stata 16.1 (StataCorp, Stata Statistical Software: Release 02. 2020, StataCorp LP: College Station, TX, USA). Risk factors were selected using a pre-specified conceptual framework with three groups of risk factors: socio-demographic characteristics, WASH access variables, and MHM practices. Guided by the Hennegan Model of menstrual experience, analysis was conducted using a hierarchal approach (Figure 1) [3,27]. Risk factors were divided into three levels. Level 1 represents the most distal risk factors and Level 3 represents the most proximate risk factors for BV and VVC. 

Datasets from Study One and Study Two were compared and variable labels and responses were renamed, regrouped, and recoded for consistency between studies. Questions that were not asked in both the Study One and Study Two survey questionnaires were dropped. Continuous variables that required grouping were recoded into categorical variables. The datasets were appended together and assessed for duplicates, internal consistency, and missing data. Each potential risk factor was cross tabulated against each outcome, BV and VVC, as each infection has different known, and potentially unknown, etiological and biological risk factors. Logistic regression was conducted to calculate the crude odds ratios (ORs) with 95% confidence intervals (CIs) for the association between each risk factor and each outcome, displayed in this analysis as (OR (95% CI)). *p*-values were calculated by the likelihood-ratio test (LRT). Tests for trend were performed with ordered categorical variables and tests for departure from linearity were carried out as appropriate. If there was no evidence of departure from linearity, the *p*-value for trend was reported. Multivariable logistic regression was conducted to determine risk factors for BV and VVC among the 509 women in the study who used reusable absorbents, adjusting for potential confounders. Multivariable analysis was conducted separately for BV and VVC. We employed a backward selection procedure with *p* < 0.15 from the univariable analysis for initial selection at each hierarchal level. Age group was retained in all final models as a priori potential confounding factor. After controlling for age group and variables retained in the same and preceding levels (adjusted OR), variables associated with the outcome at *p* < 0.1 were retained in the final model. All reported *p*-values were from the likelihood-ratio test. The hospital attended was not included a priori in the multivariable analysis as few women were recruited from Ispat General Hospital, Rourkela, and no cases of VVC were reported. However, hospital attended was added into the final multivariable models for BV to confirm that this did not result in substantial changes to the ORs and 95% CIs. Potential interacting variables were evaluated for potential effect modifiers hypothesized by the literature, including SES and household latrine privacy, by including an interaction term in the final multivariable model. The interaction-term was retained and stratum-specific odds ratios were reported if *p* < 0.05 after the likelihood-ratio test.

### 2.5. Ethical Considerations

Both cross-sectional studies were approved by the Institutional Review Board of Asian Institute of Public health (AIPH) (ERC/2013-03), Ispat General Hospital, Rourkela (Regd ECR/369/Inst/OR/2013), the Ethical Committee of Government of Odisha (237/SHRMU), and the Ethics Committee of the London School of Hygiene and Tropical Medicine (6520 and 6521). Women were only included in the study if they provided written informed consent to participate.

## 3. Results

Of the 860 reproductive-age women who visited the OG-OPD at Capital Hospital or IGH during the Study Onetime period, 106 did not consent to participate and another 196 were excluded according to our criteria (Figure 2). Of the 1023 reproductive-age women who visited the OG-OPD at Capital Hospital during the Study Two time period, 126 did not consent to participate and a further 291 were excluded according to our criteria. Of the total 1164 women enrolled in Studies One and Two, 509 (43.7%) used reusable absorbents, defined as predominately using a cloth, towel, or other absorbent material multiple times without disposal in the past six menstrual cycles, and were included in this analysis. 

### 3.1. RTI Prevalence among the Study Sample

In total, 207 (40.7%) women were diagnosed with BV, 234 (46.0%) women were diagnosed with VVC, and 76 (14.7%) women had coinfection (Figure 3). Therefore, in total, 365 (71.7%) women had at least one vaginal infection.

### 3.2. Socio-Demographic Factors

The analysis included a total of 509 women of reproductive age who visited the O&G OPD at Capital Hospital or IGH during the study intervals and who used reusable absorbents. The majority of women were recruited from Capital Hospital in Bhubaneswar (86.4%). Women in the study ranged in age from 18 to 45, with a median age of 32 years (IQR: 14). Almost all women in the study were Hindu (94.7%). Approximately half of women fell below the poverty line (54.4%), with a median of five family members per household (IQR: 2). Most women completed some formal education (65.8%), with 14.3% completing any level of higher education. The majority of women in the study were married (80.2%) and listed housewife as their primary occupation (71.1%).

The univariate analysis showed no evidence for an association between the number of family members in the household, education, religion, or marital status with BV or VVC (Table 1). Higher SES was protective against VVC, but not BV, whereby women below the poverty line had 1.5 times the odds of VVC compared to women above the poverty line (95%CI 1.0–2.1). Moreover, housewives (95%CI 1.0–2.7) and unemployed women or students (95%CI 0.9–3.5) each had 1.7 times the odds of BV compared to employed women, but there was no evidence for an association between occupation and VVC.

### 3.3. WASH Factors

Women obtained water from either inside their household (43.4%), in their yard (24.6%), or at a public place, including a relative or neighbor’s house or yard (32%). In total, 49.7% of women had access to a private household latrine, 13.2% had access to a non-private household latrine, and 37.1% did not have access to a latrine in their household. 

The univariate analysis showed no evidence for an association between the primary water source location or primary location for urination during menstruation with BV or VVC (Table 2). Household latrine privacy was associated with VVC, but not with BV. Having access to a non-private household latrine was associated with approximately double the odds of VVC compared to women without a household latrine (OR: 2.1 (1.2–3.6)); however, there was no evidence for an association between having a private household latrine and VVC (OR: 1.2 (0.8–1.7)). 

### 3.4. MHM Practices

More than half of women changed their absorbents inside of a toilet facility (52.1%), with only 22.0% changing their absorbent three or more times per day. All women reported washing themselves at least once per day. Women primarily used old cotton material (52.3%) or old silk/nylon (42.6%) as their primary absorbent during the past six cycles. More than half of women washed their absorbents inside the latrine or bathroom area (54.6%), while 45.4% washed their absorbents outside including at the tube well, public pond or river. About half of women dried their absorbents in the sun or open space (48.7%). Between cycles, women either stored absorbents in a cupboard or on shelves (52.5%) or inside the latrine or bathroom area (47.5%).

In the univariate analysis, there was no evidence for an association between the location of absorbent changing, reusable product material, or method of storing absorbents between cycles with BV or VVC (Table 3). There was no evidence for an association between absorbent changing frequency and VVC, yet frequent absorbent changing was protective against BV (*p* trend = 0.01). Washing the full body during menstruation rather than only vaginally was protective against VVC (OR: 0.6 (0.4–0.9)), but washing, either vaginally or the full body, more than twice per day was associated with 1.5 (95%CI 1.0–2.1) times greater odds of BV compared to washing once per day. Women who washed their absorbents outside in a tube well, public pond, or river had 0.7 (95%CI 0.5–0.9) times the odds of VVC than women who washed their absorbents in the latrine or bathroom area, but there was no association with BV. Moreover, drying and storage practices were associated with VVC but not BV. There was very strong evidence that drying absorbents inside the house rather than in a sunlit or open space was associated with 3.4 (95%CI 2.3–4.8) times the odds of VVC. Storing absorbents inside the latrine area was associated with increased odds of VVC (OR:1.8 (1.3–2.6)).

### 3.5. Risk Factors for BV in the Multivariable Model

In the first level of the multivariable model, after adjusting for age, housewives presented with 1.8 (95%CI 1.1–2.9) times greater odds of BV than employed women (Table 4). This did not change from the univariable analysis, suggesting age was not a confounder of the association between occupation and BV. In the second level, after adjusting for age and occupation, none of the explored WASH factors were associated with BV. In the third level, after controlling for age and the retained socio-demographic variables and MHM practices, there was still strong evidence for a protective effect of frequent absorbent changing on BV, and the ORs for changing absorbents twice per day (aOR:0.7 (0.4–1.0)) and three or more times per day (aOR: 0.5 (0.3–0.8)) remained unchanged from the univariable analysis. After adjustment, washing the body more than twice per day was still associated with 1.5 (95%CI 1.0–2.2) times the odds of BV. The odds ratios for both MHM practice-related risk factors for BV remained constant after adjustment, suggesting they were not confounded by the explored socio-demographic variables, WASH factors, or other MHM practices.

### 3.6. Risk Factors for VVC in the Multivariable Model

In the first level of the multivariable model, after adjusting for SES, increasing age was associated with increased odds of VVC whereby women 26–35 (95%CI 1.0–2.4) and 36–45 (95% CI 1.0–2.5) each had 1.6 times the odds of VVC compared to women aged 18–25 (Table 5). Moreover, there remained strong evidence of a protective effect of higher SES on VVC after controlling for age group (aOR: 1.5 (1.1–2.2)). In the second level, having a non-private household latrine was still associated with about double (aOR: 2.2 (1.3–4.0)) the odds of VVC compared to no household latrine after adjusting for age group and SES. This did not change from the univariable analysis, suggesting the association between household latrine privacy and VVC was not confounded by age or SES. In the third level, after controlling for age group and the retained sociodemographic variables, WASH factors, and MHM practices, washing absorbents outside was still protective against VVC (aOR: 0.7 (0.5–1.0)). Moreover, drying absorbents inside the house (aOR: 3.7 (2.5–5.5)) and storing absorbents inside the latrine area (aOR: 2.0 (1.3–2.9)) were still associated with increased odds of VVC. The odds ratios for all MHM practice-related risk factors for VVC remained constant after adjustment, suggesting they were not confounded by the explored socio-demographic variables, WASH factors, or other MHM practices. 

In terms of interaction in the final model, there was no evidence for a difference in the effect of household latrine privacy on VVC among women above compared to below the poverty line (*p* = 0.08), thus the stratum specific ORs were not reported.

## 4. Discussion

The current study supports the hypothesis that certain socio-demographic characteristics, WASH factors, and MHM practices among Odisha women using reusable menstrual products are associated with a higher risk of lower RTIs. As expected, we found different factors were associated with BV and with VVC. After adjusting for confounding variables using a hierarchal framework, frequent absorbent changing was protective against BV, whereas frequent body washing increased the odds of BV. Women with VVC were more likely to be older, below the poverty line, have a non-private household latrine, dry absorbents inside the house, and store absorbents in the latrine area. Washing absorbents outside in the tube well, public pond, or river was protective against VVC.

Among the 509 women enrolled in the study who used reusable absorbents, the RTI prevalence was 71.7%, with a prevalence of 40.7% for BV and 46.0% for VVC. This RTI prevalence is approximately double the 35.2% RTI/STI prevalence reported in the DLHS-RCH (2002–2004) survey in Odisha [22]. This increase may be due to the high prevalence of asymptomatic cases and the syndromic reporting of RTIs in the DLHS-RCH survey. It may also be due to the study population, as it is plausible that women attending the O&G unit would have higher rates of infection compared to the wider Odisha population. Nevertheless, the prevalence observed in our study was similar to those of other Indian studies [28,29,30].

Being employed was protective against BV. Previous qualitative research in Odisha suggests that the type of occupation is a determinant of latrine use and open defecation [31]. For example, farmers and people who work outdoors far away from a latrine reported the inconvenience associated with latrine use and were more likely to practice open defecation. However, people who worked in hostels or government buildings were more compelled to use latrine facilities at work [31]. Likewise, it is plausible that the type and location of employment, in terms of latrine access and availability, privacy, and opportunities to change absorbents and wash the body at work during menstruation, may all be factored in to determine the risk of BV. Thus, although our study found that employment was protective against BV, it is also plausible that certain types of employment may increase the risk of BV. Lower SES was associated with increased risk of VVC, but surprisingly not BV. This remained constant in the multivariable model including age, WASH factors, and MHM practices, suggesting an independent effect of SES on VVC infection. Investigating associations with SES showed that women below the poverty line were more likely to dry absorbents inside the house, which may partly explain their increased risk of VVC (Appendix A). It is also plausible that there are other unexplored mechanisms for the effect of SES on VVC, as women in similar contexts report challenges to managing menstruation in resource-poor settings, such as acquiring soap and water [31,32]. 

In line with previous research, women with a non-private household latrine had an increased risk of VVC compared to women without a household latrine, after adjusting for socio-demographic factors. Previous research has found that lack of latrine privacy was associated with increased psychosocial stress, shame, embarrassment, and harassment, which may explain some of this association [32,33]. There was no evidence, however, for an association between having a private household latrine and VVC. One reason for the lack of evidence may be the variability of household latrines in terms of infrastructure, use, and conditions. Due to the hospital-based case–control study design, we were not able to travel to the community and do spot checks to assess the latrines conditions and other WASH factors.

Consistent with previous research in Odisha, women who changed their absorbents more frequently had a lower risk of BV [7]. The exact biological mechanism for how changing absorbents is related to BV is unknown. However, BV is caused by an imbalance of vaginal flora, which is known to be influenced by several factors, such as sexual activity, douching, and host immune response [7,11,34]. These known biological mechanisms support the presumption that the vaginal ecosystem may be influenced by the accumulation of blood in the vagina for prolonged periods of time [7]. Interestingly, increased frequency of bathing during menstruation was also associated with an increased risk of BV. This suggests that washing more frequently may also alter the vaginal flora. However, this finding conflicts with a previous study in the same setting, which found that increased bathing practices were associated with a lower risk of BV [25]. One possible reason for this inconsistency in findings is the variation in diagnostic techniques for detecting BV, as the previous study used Amsel’s clinical diagnostic criteria for diagnosis, which is less sensitive than Nugent’s criteria used in this study [7,25,35]. Additionally, no women in our study reported washing less than once per day during menstruation, which suggests the possibility of social desirability bias, leading to an over-estimation of bathing practices. 

Absorbent washing, drying, and storage were all strongly associated with VVC. Washing absorbents in the tube well, public pond, or river was associated with lower odds of VVC than washing absorbents in a latrine or bathroom facility. We hypothesize that this association varies depending on the infrastructure of the latrine facility, privacy, and the availability of clean water and soap. Although all women reported washing their absorbents with soap or detergent, this may have been partially due to social desirability bias as women in a previous qualitative study in Odisha reported the challenges women often face obtaining soap [32]. Moreover, consistent with previous research, drying absorbents in the sun or open space and storing absorbents in the cupboard or on shelves were protective against VVC [1,8]. Storing damp absorbents has been found to promote microbial survival and wearing absorbents that are not fully dry may lead to an abnormally moist environment in the vulvovaginal area, subsequently promoting the growth of *Candida albicans* and the development of VVC [7,25,36,37]. Although there is robust evidence and widespread knowledge for the protective effect of drying absorbents outside in the sun, previous qualitative studies in Odisha highlighted the challenges associated with storing absorbents hygienically and drying absorbents outside [32,38]. These barriers predominantly relate to the stigma, shame, and embarrassment women felt at the prospect of men seeing their absorbents [32].

There are several limitations of this study. Given that this is an observational, cross-sectional study, we cannot infer causality based on the observed associations between risk factors and infections. Among the 1164 women enrolled in the original two studies, only 509 used reusable pads and were included in this analysis. As a result of the relatively small sample size, there was limited power to detect a significant association between certain risk factors and BV or VVC. The survey questionnaires used in the two studies were slightly different, limiting our ability to explore potentially important risk factors for BV and VVC that were not included in both studies. Some answer choices were recoded for consistency, resulting in larger groupings of responses to accommodate both sets of answer choices. Additionally, only women with access to a household latrine were prompted to answer questions about WASH access and infrastructure of their latrine facility, consequently excluding many WASH variables from the analysis. Moreover, this was a hospital-based study and only women attending the obstetrics and gynecology departments consented to participate in the study, and to provide vaginal swabs, such that our findings may not be representative of the wider population attending other sections of the hospital, thus limiting the generalizability of our findings to the wider population. Furthermore, VVC was diagnosed by the AlbiQuick^TM^ rapid test, which detects the presence of *Candida albicans*, which causes 85% to 90% of VVC [39]. However, VVC can also be caused by other *Candida* species or yeasts, which cannot be detected by the AlbiQuick^TM^ rapid test. Subsequently, VVC may have been underdiagnosed, potentially leading to a dilution of the strength of association for some risk factors and VVC. Moreover, questions regarding other potential risk factors for RTIs, such as sexual practices, were not explored due to the strong stigma against women speaking about sexual practices in Odisha. Thus, they were not adjusted for in the analysis. Although we adjusted for a wide range of potential confounders at each level of the hierarchal framework, residual confounding likely remains.

## 5. Conclusions

The results from this study demonstrate a strong association between various WASH and MHM practices and two lower RTIs, BV and VVC, among Odisha women reusing menstrual absorbents. Due to the high use of reusable absorbents in Odisha, nearly 44% of women in our study, our findings further highlight the need for improved MHM practices related to changing, washing, drying, and storage of absorbents. Our study supports global action for the provision of menstrual hygiene promotion programs that offer advice to women about how to change, wash, dry, and store their absorbents with dignity. Future efforts should seek to evaluate the impact of these interventions on reducing the prevalence of lower RTIs. More research in India and other parts of the world is also needed to further investigate the associations between WASH conditions and lower RTIs, with a larger sample size and alternative study design, allowing for spot checks and further exploration of specific WASH conditions. 

## Figures and Tables

**Figure 1 ijerph-18-04778-f001:**
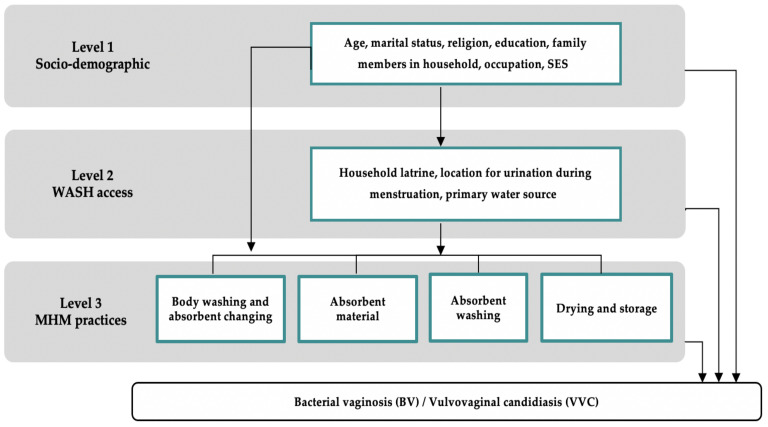
Hierarchal conceptual framework.

**Figure 2 ijerph-18-04778-f002:**
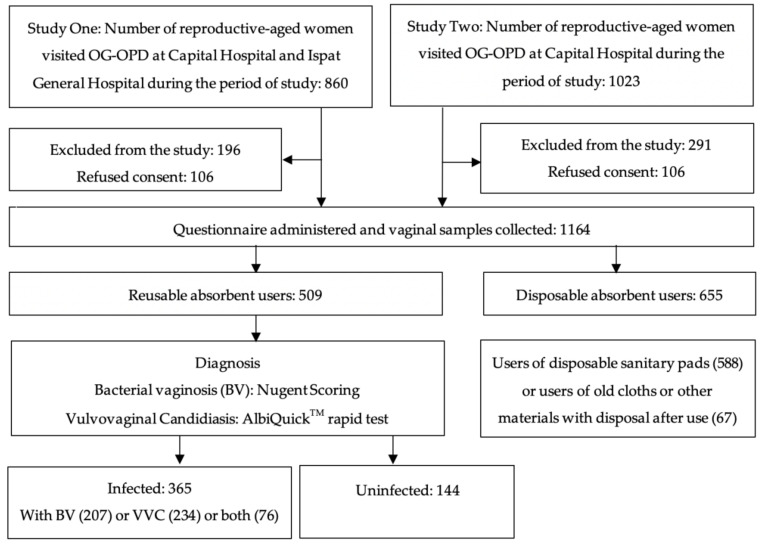
Schematic diagram for recruitment, enrollment, analysis inclusion, outcome diagnosis, and inclusion in the analysis.

**Figure 3 ijerph-18-04778-f003:**
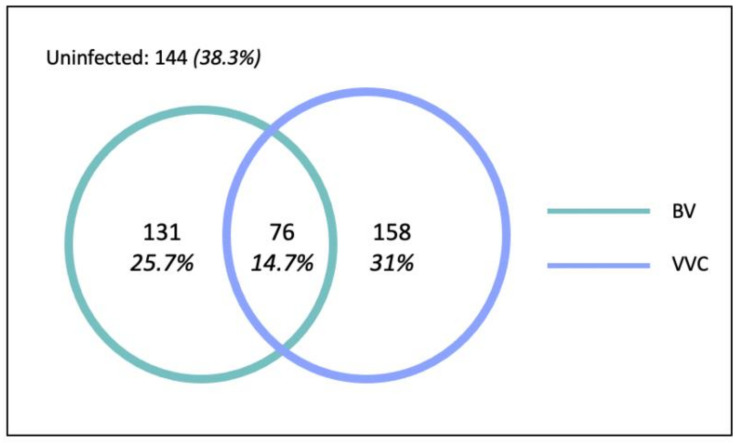
Study population by outcome (N = 509). Circles are proportionate to sample size.

**Table 1 ijerph-18-04778-t001:** Crude odds ratios for BV and VVC according to socio-demographic factors (N = 509).

Level 1 Factors	N	BV+	Crude OR	*p*-Value (LRT)	VVC+	Crude OR	*p*-Value (LRT)
		(%)	(95% CI)		(%)	(95% CI)	
Hospital attended				0.99			-
Capital Hospital, Bhubaneswar	440	221 (43.6)	1		243 (47.9)	1	
Ispat General Hospital, Rourkela	69	28 (40.6)	1.0 (0.6–1.7)		0	-	
Age				0.45			0.14
18–25	141	63 (44.7)	1		55 (39.0)	1	
26–35	188	76 (40.4)	0.8 (0.5–1.3)		90 (47.9)	1.4 (0.9–2.2)	
36–45	180	68 (37.8)	0.8 (0.5–1.2)		89 (49.4)	1.5 (1.0–2.4)	
Family members in household				0.94			0.61
1–2	74	29 (39.2)	1		37 (50.0)	1	
4–5	291	120 (41.2)	1.1 (0.6–1.8)		135 (46.4)	0.9 (0.5–1.4)	
6+	144	58 (40.3)	1.0 (0.6–1.9)		62 (43.1)	0.8 (0.4–1.3)	
Below Poverty Line				0.51			0.04
No	232	98 (42.2)	1		95 (41.0)	1	
Yes	277	109 (39.4)	0.9 (0.6–1.3)		139 (50.2)	1.5 (1.0–2.1)	
Education				0.79			0.24
No formal education	101	42 (41.6)	1		39 (38.6)	1	
Some formal education ^1^	335	133 (39.7)	0.9 (0.6–1.5)		161 (48.1)	1.5 (0.9–2.3)	
Any level of higher education	73	32 (43.8)	1.1 (0.6–2.0)		34 (46.6)	1.4 (0.8–2.6)	
Occupation				0.09			0.27
Employed	94	29 (30.9)	1		50 (53.2)	1	
Housewife	362	155 (42.8)	1.7 (1.0–2.7)		162 (44.8)	0.7 (0.5–1.1)	
Unemployed/student	53	23 (43.4)	1.7 (0.9–3.5)		22 (41.5)	0.6 (0.3–1.2)	
Religion				0.22			0.53
Hindu	482	199 (41.3)	1		220 (45.6)	1	
Muslim/Christian	27	8 (29.6)	0.6 (0.3–1.4)		14 (51.9)	1.3 (0.6–2.8)	
Marital status				0.2			0.58
Single ^2^	76	31 (40.8)	1		35 (46.1)	1	
Married	408	170 (41.7)	1.0 (0.6–1.7)		185 (45.3)	1.0 (0.6–1.6)	
Widowed/divorced	25	6 (24.0)	0.5 (0.2–1.3)		14 (56.0)	1.5 (0.6–3.7)	

^1^ Completed primary or secondary education; ^2^ Never married.

**Table 2 ijerph-18-04778-t002:** Odds ratios for BV and VVC according to WASH factors (N = 509).

Level 2 Factors	N	BV+	Crude OR	*p*-Value (LRT)	VVC+	Crude OR	*p*-Value (LRT)
		(%)	(95% CI)		(%)	(95% CI)	
Availability of a latrine in the household				0.67			0.04
No	189	73 (38.6)	1		79 (41.8)	1	
Yes, non-private	67	30 (44.8)	1.3 (0.7–2.3)		40 (59.7)	2.1 (1.2–3.6)	
Yes, private	253	104 (41.1)	1.1 (0.8–1.6)		115 (45.5)	1.2 (0.8–1.7)	
Primary water source location				0.35			0.62
In the house	221	96 (43.4)	1		99 (44.8)	1	
In the yard	125	52 (41.6)	0.9 (0.6–1.4)		55 (44.0)	1.0 (0.6–1.5)	
Public location ^1^	253	59 (36.2)	0.7 (0.5–1.1)		80 (49.1)	1.2 (0.8–1.8)	
Primary location for urination during menstruation				0.78			0.27
Outside of house/yard	198	82 (41.4)	1		85 (42.9)	1	
Latrine inside house/yard	311	125 (40.2)	1.0 (0.7–1.4)		149 (47.9)	1.2 (0.9–1.8)	

^1^ Including a relative or neighbor’s house or yard.

**Table 3 ijerph-18-04778-t003:** Odds ratios for BV and VVC according to MHM practices (N = 509).

Level 3 Factors	N	BV+	Crude OR	*p*-Value (LRT)	VVC+	Crude OR	*p*-Value (LRT)
		(%)	(95% CI)		(%)	(95% CI)	
**Changing absorbents or washing the body**							
*Location of absorbent changing*				0.39			0.70
Inside toilet facility	265	103 (38.9)	1		124 (46.8)	1	
Outside of toilet facility	244	104 (42.6)	1.2 (0.8–1.7)		110 (45.1)	0.9 (0.7–1.3)	
*Frequency of absorbent changing on heavier days*				0.01 ^1^			0.62
Once/day	159	76 (47.8)	1		74 (46.5)	1	
Twice/day	238	94 (39.5)	0.7 (0.5–1.1)		113 (47.5)	1.0 (0.7–1.6)	
Three or more times/day	112	37 (33.0)	0.5 (0.3–0.9)		47 (42.0)	0.8 (0.5–1.4)	
*Type of body washing during menstruation*				0.34			0.02
Vaginal wash only	160	70 (43.8)	1		86 (53.8)	1	
Full body bath	349	137 (39.3)	0.8 (0.6–1.2)		148 (42.4)	0.6 (0.4–0.9)	
*Frequency of washing during menstruation* ^2^				0.05			0.88
Once per day	289	106 (36.7)	1		132 (45.7)	1	
Twice or more per day	220	101 (45.9)	1.5 (1.0–2.1)		102 (46.4)	1.0 (0.7–1.5)	
**Material**							
*Reusable product material*				0.47			0.99
Old cotton (sari or other)	266	102 (38.4)	1		122 (45.9)	1	
Old silk/nylon (sari or other)	217	95 (43.78)	1.3 (0.9–1.8)		100 (46.1)	1.0 (0.7–1.4)	
Towel	26	10 (38.5)	1.0 (0.4–2.3)		12 (46.2)	1.0 (0.5–2.3)	
**Absorbent washing**							
*Location of washing reusable absorbent*				0.46			0.02
Latrine/bathroom area	278	109 (39.2)	1		141 (50.7)	1	
Outside ^3^	231	98 (42.4)	1.1 (0.8–1.6)		93 (40.3)	0.7 (0.5–0.9)	
**Drying and storage**							
*Location of drying absorbents*				0.16			<0.01
Sun or open space	248	93 (37.5)	1		77 (31.1)	1	
Inside the house	261	114 (43.7)	1.3 (0.9–1.8)		157 (60.2)	3.4 (2.3–4.8)	
*Method of storing absorbents between cycles*				0.96			0.96
Wrapped ^4^	415	169 (40.7)	1		191 (46.0)	1	
Without wrapping	94	38 (40.4)	1.0 (0.6–1.6)		43 (45.7)	1.0 (0.6–1.5)	
*Location of absorbent storage between cycles*				0.94			<0.01
Cupboard/shelves	267	109 (40.8)	1		104 (39.0)	1	
Latrine or bathroom	242	98 (40.5)	1.0 (0.7–1.4)		130 (53.7)	1.8 (1.3–2.6)	

^1^*p*-value from the test for trend; ^2^ Vaginally or full body bathing; ^3^ Tube well, public pond, or river; ^4^ Wrapped in polythene, paper, or in a container.

**Table 4 ijerph-18-04778-t004:** Final hierarchical model for BV. Odds ratios are adjusted for variables in the same and preceding levels. Age group was included a priori confounding factor in the models at every level.

Variable	Adjusted OR (95% CI)
**Level 1**	
Age	
18–25	1
26–35	0.8 (0.5–1.2)
36–45	0.7 (0.4–1.1)
Occupation	
Employed	1
Housewife	1.8 (1.1–2.9)
Unemployed/student	1.4 (0.7–3.0)
**Level 2** No variables retained	
**Level 3**	
Absorbent changing frequency on heavier days	
Once per day	1
Twice per day	0.7 (0.4–1.0)
Three+ times per day	0.5 (0.3–0.8)
Body washing frequency during menstruation	
Once per day	1
Twice or more per day	1.5 (1.0–2.2)

**Table 5 ijerph-18-04778-t005:** Final hierarchical model for VVC. Values represent odds ratios with confidence intervals in parentheses. Odds ratios are adjusted for variables in the same and preceding levels. Age group was included a priori confounding factor in the models at every level.

Variable	Adjusted OR (95% CI)
**Level 1**	
Age	
18–25	1
26–35	1.6 (1.0–2.4)
36–45	1.6 (1.0–2.5)
Below Poverty Line	
No	1
Yes	1.5 (1.1–2.2)
**Level 2**	
Availability of a latrine in the household	
No	1
Yes, without privacy	2.2 (1.3–4.0)
Yes, with privacy	1.3 (0.9–1.9)
**Level 3**	
Absorbent washing	
Latrine area	1
Outside ^1^	0.7 (0.4–1.0)
Absorbent drying	
Sunlit/open space	1
Inside house	3.7 (2.5–5.5)
Absorbent storage	
Cupboard	1
Latrine area ^2^	2.0 (1.3–2.9)

^1^ Washing absorbents outside includes in a tube well, public pond, or river. ^2^ Women reported storing the absorbent under the roof or wall of the latrine area.

## Data Availability

The data presented in this study are available on reasonable request from the corresponding author.

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
