# Peer review of "Identifying Risk Factors for Lower Reproductive Tract Infections among Women Using Reusable Absorbents in Odisha, India"

_ijerph, 2021, doi:10.3390/ijerph18094778_

Round 1

Reviewer 1 Report

The authors examined specific risk factors related to MHM practices among reproductive aged women in Odisha who use reusable menstrual products and their associations with the most common RTIs: BV and VVC. 

This is an important study as it highlights the improved MHM practices related to changing, washing, drying, and storage of absorbents

The authors mention A large proportion of women in Odisha, India use reusable absorbents to manage their menstruation however, the risk factors for lower reproductive tract infections related to menstrual hygiene management (MHM) have not been studied among reusable absorbent users.

This study adds to the area of research and has compelling scientific evidence to support their findings. The study use Nugent scoring for BV diagnosis while other studies mentions uses Amsel's criteria which maybe one of the reasons they found significant associations.

The paper is well written. The conclusions are consistent with published literature. The study has compelling evidence to support their finding and is performed on two large cohort

Specific comments:

Line 56: Mention the types of lactobacillus sp. as some are more important. Lactobacillus crispatus, Lactobacillus jensenii, Lactobacillus iners.
Line 57: Mention the most important, such as Gardnerella vaginalis, Prevotella bivia, BV-associated-bacteria-2 (BVAB2), Megasphaera 1, Atopobium vaginae  etc.
Line 61: Mention BV increase your risk to HIV-1 infection
Line 138: Mention BV scoring was performed by trained peroneal 
Line 139: BV 0-3 NS, BV 4-7 intermediate 7-10 BV positive, should mention this here and the organisms seen. The study uses 4 -10 as BV positive and it will help if the authors mentioned if these patients were referred for BV treatment.

Author Response

Reviewer 1:

The authors examined specific risk factors related to MHM practices among reproductive aged women in Odisha who use reusable menstrual products and their associations with the most common RTIs: BV and VVC. 

This is an important study as it highlights the improved MHM practices related to changing, washing, drying, and storage of absorbents

The authors mention A large proportion of women in Odisha, India use reusable absorbents to manage their menstruation however, the risk factors for lower reproductive tract infections related to menstrual hygiene management (MHM) have not been studied among reusable absorbent users.

This study adds to the area of research and has compelling scientific evidence to support their findings. The study use Nugent scoring for BV diagnosis while other studies mentions uses Amsel's criteria which maybe one of the reasons they found significant associations.

The paper is well written. The conclusions are consistent with published literature. The study has compelling evidence to support their finding and is performed on two large cohort

Specific comments:

Line 56: Mention the types of lactobacillus sp. as some are more important. Lactobacillus crispatus, Lactobacillus jensenii, Lactobacillus iners.

Answer: Change added in lines 58-9.

Line 57: Mention the most important, such as Gardnerella vaginalis, Prevotella bivia, BV-associated-bacteria-2 (BVAB2), Megasphaera 1, Atopobium vaginae  etc.

Answer: Change added in line 60-61.

Line 61: Mention BV increase your risk to HIV-1 infection.

Answer: Change added in lines 64-5.

Line 138: Mention BV scoring was performed by trained peroneal 

Answer: Change added in lines 141-2.

Line 139: BV 0-3 NS, BV 4-7 intermediate 7-10 BV positive, should mention this here and the organisms seen. The study uses 4 -10 as BV positive and it will help if the authors mentioned if these patients were referred for BV treatment.

Answer: Clarifications about how we assessed Nugent score and other changes suggested by the reviewer have been added in lines 143-150.

Reviewer 2 Report

This is a well conducted study in terms of various aspects of methodology, where authors through questionnaires collected information on MHM practices, water, sanitation, and socio-demographic factors, studying 509 women who used reusable absorbents. Authors conclude that their study highlights the importance of improving MHM practices among reusable absorbent users to prevent lower RTIs infection.

My main concern is that this study was performed in a specific population in a specific town in India and although the methodology was appropriate the final results cannot merit generalisation. My secondary concern is that I wonder what the unknown observed through the study clinical message might be.

I would propose the addition of a comparator, in order all this effort to gain more sense (of another testing for example or another study population). The fact that this is a part from a bigger study justifies this gap. The reporting in all sections could be shortened. In the discussion section direct comparisons with similar reports should be enriched.

Author Response

Reviewer 2:

This is a well conducted study in terms of various aspects of methodology, where authors through questionnaires collected information on MHM practices, water, sanitation, and socio-demographic factors, studying 509 women who used reusable absorbents. Authors conclude that their study highlights the importance of improving MHM practices among reusable absorbent users to prevent lower RTIs infection.

My main concern is that this study was performed in a specific population in a specific town in India and although the methodology was appropriate the final results cannot merit generalisation. My secondary concern is that I wonder what the unknown observed through the study clinical message might be. I would propose the addition of a comparator, in order all this effort to gain more sense (of another testing for example or another study population). The fact that this is a part from a bigger study justifies this gap.

Answer: We appreciate your comments, but we would like to clarify that the study was conducted in two study sites of Odisha (Capital Hospital in Bhubaneswar and Ispat General Hospital (IGH) in Rourkela). Both sites are tier-2 cities in Odisha, especially Bhubaneswar which is the capital cities of this state). The capture area of both tertiary health centres is comprised of women living in these cities, and women coming from different villages from Odisha state, including tribal communities in the area of Rourkela. These details are described between lines 112-119. We also recognized that this study has only been done in hospital setting (due to the complication of collection of samples and technical laboratory analysis of them in other settings), and we acknowledge that it is more complicate to generalise results from clinical based than population based. But we also consider that the strengths in our work include a good sample size, the use of an expert microbiological laboratory with external quality controls to diagnose BV and Candida infection and the fact that all of the doctors and interviewers involved were females, which assured a relaxed environment to assess exposure factors and discuss a stigmatized topic. We have added few points to discuss further generalisability in lines 426-430, and also we have made few changes in the abstract and conclusion, to clarify that these results are only representing Odisha women and that further research in other parts of India and the world are also needed.  The reviewer suggests the use of other comparator (eg. Another study population), we would like to clarify again that this study was conducted in two different sites of Odisha, and also that the study ended up few years ago, and we are not able to repeat another full study.

The reporting in all sections could be shortened. In the discussion section direct comparisons with similar reports should be enriched.

Answer: We appreciate your comments, we have shortened Introduction and some results sections, but we have not made changes in method sections, as we believe that all the information presented is needed for the readers, to have a clear understanding of the study design and methods.  We have added more information about similar reports that support the findings of our study along introduction and discussion sections.

Round 2

Reviewer 2 Report

Authors made a very important effort to comply with suggestions